# Psychological, Relational, and Fertility-Related Characteristics of Italian Women with Vulvodynia: A Comparative Study with Controls

**DOI:** 10.3390/ijerph22040527

**Published:** 2025-03-30

**Authors:** Antonio Gattamelata, Giulia Fioravanti, Vanessa Prisca Zurkirch, Nieves Moyano

**Affiliations:** 1Escuela de Doctorado en Psicología, Universidad de Jaén, Campus Las Lagunillas s/n, 23071 Jaén, Spain; mnmoyano@ujaen.es; 2Department of Psychology, Universidad de Jaén, Campus Las Lagunillas s/n, Ed.C5, 23071 Jaén, Spain; 3Department of Health Sciences (DSS), University of Florence, 50135 Florence, Italy; giulia.fioravanti@unifi.it; 4Maternal and Child Department, Regional Reference Center on Relational Criticalities (RCRC), Careggi University Hospital, 50134 Florence, Italy; vanessaprisca.zurkirch@unifi.it

**Keywords:** vulvodynia, psychological distress, attachment styles, resilience, fertility distress, sexual function, dyadic adjustment, integrated interventions

## Abstract

Vulvodynia, a chronic pain condition affecting the vulvar area, is associated with significant psychological distress and profoundly impacts women’s quality of life. This study examines the psychological and relational experiences of women with vulvodynia, focusing on attachment styles, resilience, fertility-related distress, and sexual functioning. A total of 203 women (96 with vulvodynia and 107 controls), aged 19 to 50 years, were recruited and completed a series of validated psychological measures. The results revealed that women with vulvodynia reported higher levels of attachment anxiety and avoidance, lower resilience, greater fertility-related distress, impaired sexual function, reduced dyadic adjustment, and elevated depressive symptoms compared to the controls. These findings underscore the complex interplay of psychological, relational, and fertility-related challenges experienced by women with vulvodynia. This study highlights the need for integrated, multidisciplinary approaches to address the medical, psychological, and relational dimensions of the condition, ultimately improving the well-being and quality of life for affected women.

## 1. Introduction

Vulvodynia, a chronic condition characterized by persistent vulvar pain without an identifiable cause, affects approximately 12–15% of women in Italy [1,2,3]. The International Society for the Study of Vulvar Diseases (ISSVD) defines it as “chronic discomfort involving the vulva without an obvious etiology” persisting for at least three months. The etiology of vulvodynia is multifactorial, potentially involving local injury or inflammation, and peripheral or central sensitization of the nervous system [4,5]. Studies suggest that women with vulvodynia experience significant disruptions in sexual, relational, and psychological functioning, which can be as distressing as the physical pain itself [6,7,8,9,10,11]. Moreover, vulvodynia can introduce significant challenges for women attempting to conceive. The associated dyspareunia (painful intercourse) may lead to reduced sexual activity, potentially complicating conception efforts [12,13]. Additionally, the psychological stress and anxiety stemming from persistent pain can further impact reproductive planning and sexual relationships. Despite its prevalence, vulvodynia remains underdiagnosed and poorly understood, often overlooked due to factors such as limited awareness and diagnostic challenges [14,15].

This study focuses on Italian women, highlighting the psychological, relational, and fertility-related challenges they face after a diagnosis of vulvodynia. Until recently in Italy, vulvodynia was not acknowledged as a pathology within the Essential Levels of Care (LEAs), which restricted access to appropriate treatments. Furthermore, vulvar pain is frequently viewed as a “taboo” topic in Italian society, often deterring women from seeking help due to fears of being misunderstood or not taken seriously [16,17,18].

The primary aim of this study was to explore the psychological, relational, and fertility-related characteristics of women with vulvodynia. Specifically, we sought to compare these dimensions between women diagnosed with vulvodynia and a control group without chronic pain conditions. While previous research has examined risk and protective factors in the development of vulvodynia [19,20,21], our focus was on post-diagnostic characteristics, including attachment styles, resilience, sexual functioning, and fertility-related distress.

Understanding these dimensions is essential for developing integrated care strategies that address both the medical and psychosocial aspects of vulvodynia. Focusing on post-diagnostic characteristics allows for the creation of comprehensive interventions that tackle the diverse challenges faced by affected women. By exploring these factors, this study aims to illuminate the lived experiences of women with vulvodynia, informing care models that prioritize holistic patient support.

## 2. Materials and Methods

### 2.1. Participants

After signing the informed consent form, 214 Italian-speaking subjects participated in this study and completed the battery test.

Participants were recruited using a combination of online and direct outreach methods:-Online Recruitment: Invitations were shared on social media platforms, forums, and through email campaigns.-Direct Contact: The researchers approached participants at universities, community centers, and public events, providing them with information about this study and access to the survey. The survey complied with data protection regulations and took approximately 30 min to complete. All participants received no payment and were free to leave this study at any time. The inclusion criteria were as follows: (1) being between 18 and 50 years old; (2) having the ability to read and write the Italian language; and (3) having a diagnosis of vulvodynia confirmed by a gynecologist, for the vulvodynia group, or (4) having no history of chronic pain conditions, for the control group. The exclusion criteria were as follows: (1) consumption of addictive substances (alcohol and drugs) and (2) pre-existing medical conditions. For the control group, all participants with a history of chronic pain were excluded, including conditions such as chronic back pain, fibromyalgia, migraines, or other gynecological or obstetric pathologies. This requirement was verified through a detailed sociodemographic questionnaire administered during the initial screening phase. It is important to note that participants with occasional or transient pain, but not classifiable as chronic pain (i.e., persisting for at least three months), were not excluded from this study. Considering the voluntary nature of participation in this study and the inclusion and exclusion criteria, the final sample includes 203 women, divided into two groups: 96 women (47.3%) with vulvodynia (Group 1, average age 31.9 years, SD =7.31) and 107 women (52.7%) from the general population (Group 2, average age 32.7, SD = 7.32).-The diagnosis of vulvodynia was confirmed by a qualified gynecologist following the standardized criteria established by the International Society for the Study of Vulvovaginal Diseases (ISSVD) [1]. According to these criteria, vulvodynia is defined as persistent vulvar pain without an identifiable cause, lasting for at least three months. To be included in this study, participants were required to have received a formal diagnosis of vulvodynia, verified through medical documentation or confirmation letters issued by their healthcare providers. This approach aimed to reduce heterogeneity in the diagnostic process and minimize the risk of selection bias.-To further reduce the risk of diagnostic heterogeneity, participants were primarily recruited from Italian regions where specific Diagnostic Therapeutic Care Pathways (PDTAs) for vulvodynia have been implemented, such as those active in Tuscany. These pathways aim to standardize diagnostic procedures and ensure timely and accurate diagnoses.

Table 1 presents the sociodemographic information in detail.

### 2.2. Measures

The participants completed a series of questionnaires, including sociodemographic information and the following psychometric scales, all validated in the Italian language:-Experience in Close Relationship Scale 12 (ECR-12) [22]: The ECR-12 is a 12-item self-report instrument designed to assess adult attachment patterns in romantic relationships. It evaluates two dimensions: attachment avoidance and attachment anxiety. The scale has demonstrated good psychometric properties and is widely used in psychotherapy research. Cronbach’s alpha for the ECR-12 in this study was 0.87.-Approach-Avoidance Temperament Questionnaire (ATQ) [23]: The ATQ is a psychometric tool developed to measure individual differences in approach and avoidance temperament. It assesses tendencies toward positive (approach) and negative (avoidance) emotional responses, which are fundamental aspects of personality influencing behavior and emotional experiences. Cronbach’s alpha for the ATQ was 0.82.-Resilience Scale 14 (RS-14) [24]: The RS-14 is a 14-item scale designed to measure resilience, defined as the ability to cope with and adapt to adversity. It provides insights into an individual’s capacity to bounce back from stressful situations and maintain psychological well-being. Cronbach’s alpha for the RS-14 was 0.91.-Fertility Problem Inventory—Short Form (FPI-SF) [25]: The FPI-SF is a shortened version of the Fertility Problem Inventory, aimed at assessing the perceived stress and impact associated with fertility problems. It evaluates various dimensions, including social, sexual, and relational aspects, providing a comprehensive understanding of the psychological burden of infertility. Cronbach’s alpha for the FPI-SF was 0.85.-Female Sexual Function Index (FSFI) [26]: The FSFI is a 19-item self-report questionnaire designed to measure sexual functioning in women. It assesses six domains: sexual desire, arousal, lubrication, orgasm, satisfaction, and pain. The FSFI is widely used in both clinical and research settings to evaluate female sexual function and identify potential dysfunctions. Cronbach’s alpha for the FSFI was 0.94.-Dyadic Adjustment Scale 7 (DAS-7) [27]: The DAS-7 is a brief version of the Dyadic Adjustment Scale, consisting of 7 items. It measures the quality of adjustment and satisfaction within a romantic relationship, covering aspects such as consensus, cohesion, and satisfaction between partners. Cronbach’s alpha for the DAS-7 was 0.78.-Beck Depression Inventory (BDI) [28]: The BDI is a 21-item self-report inventory that assesses the presence and severity of depressive symptoms. It covers a range of emotional, cognitive, and physical symptoms associated with depression and is one of the most widely used instruments for detecting and quantifying depressive states. Cronbach’s alpha for the BDI was 0.89.

### 2.3. Procedure

Data collection took place between March 2024 and February 2025. The participants were recruited through a dedicated website (www.psicologia-pma.com, accessed on 1 March 2024), which hosted an online version of the test battery, or via email invitations requesting their participation in this study. Before beginning the survey, participants received a brief, standardized explanation from an interviewer to ensure informed consent and a clear understanding of the study objectives. Although the survey was self-administered, the interviewer was available to address questions and improve response accuracy, particularly for participants less familiar with such assessments.

Participants completed the survey in approximately 30 min, which included the psychometric test battery and a sociodemographic questionnaire. Regarding research standards, this study adhered to the most recent version of the Declaration of Helsinki (WMA, 2013). This study was approved by the University Ethics Committee of the University of Jaén (protocol code, ABR.23/17 TES; approval date, 28 April 2023).

### 2.4. Statistical Analysis

#### 2.4.1. Data Preprocessing and Exclusion Criteria

To ensure data quality, the following preprocessing steps were adopted:

Participants with more than 20% missing data were excluded from the analysis (*n* = 11).

Remaining missing values were imputed using multiple imputation techniques.

Outliers were identified and handled using the interquartile range (IQR) method.

#### 2.4.2. Data Processing

In the first step of the analysis, the item values were examined in order to analyze the frequency and variance distributions. Descriptive statistics (means, standard deviations, and percentages) were computed for all sociodemographic and clinical variables.

The assumptions of normality and homogeneity of variance were verified for all continuous variables. The Shapiro–Wilk test confirmed that most variables met the assumption of normality. For variables with slight deviations from normality, logarithmic transformations were applied to normalize the distribution. Additionally, Levene’s test confirmed the homogeneity of variance for all comparisons.

To compare sociodemographic characteristics between the two groups (vulvodynia vs. control), chi-square tests (χ^2^) were performed for categorical variables, and a t-test was conducted for the continuous variable. Effect sizes (Cohen D) were calculated for each t-test to assess the magnitude of group differences. All statistical tests were two-tailed, and the significance level was set at *p* < 0.05.

Data analyses were conducted using the Statistical Package for Social Sciences (SPSS 28.0; IBM Corp., Armonk, NY, USA).

## 3. Results

### 3.1. Sociodemographic Variables

The chi-square analyses revealed significant differences between the vulvodynia group and the control group in marital status (χ^2^(5) = 15.892, *p* = 0.007) and relationship type (χ^2^(6) = 15.467, *p* = 0.017). Women with vulvodynia were more likely to be separated or divorced, while controls were more frequently cohabiting with a partner or engaged. No significant differences emerged in other sociodemographic variables such as education level, employment status, or income level (*p* > 0.05). Table 1 presents the sociodemographic characteristics of both groups.

### 3.2. Clinical Variables

The analysis of clinical variables revealed significant differences between the vulvodynia group and the control group across most measures (Table 2). Notably, women with vulvodynia reported higher levels of infertility-related distress compared to the controls, as evidenced by the Fertility Problem Inventory—Short Form (FPI-SF) subscales. However, no significant differences were observed in the subscales of Need for Parenthood and Rejection of a Child-Free Lifestyle.

Women with vulvodynia exhibited markedly higher levels of sexual dysfunction, as measured by the Female Sexual Function Index (FSFI total score: t(198) = 6.96, *p* < 0.001, Cohen’s d = 0.98). This dysfunction was evident across all domains of sexual function, including desire, arousal, lubrication, orgasm, pain, and satisfaction. Similarly, depressive symptoms were significantly more pronounced in the vulvodynia group (BDI total score: t(199) = 15.04, *p* < 0.001, Cohen’s d = 2.22), reflecting the substantial psychological burden associated with the condition.

Significant group differences were also observed in attachment-related measures. Women with vulvodynia reported higher levels of attachment anxiety (ECR—Anxiety: t(198) = 6.96, *p* < 0.001, Cohen’s d = 0.98) and attachment avoidance (ECR—Avoidance: t(200) = 14.27, *p* < 0.001, Cohen’s d = 2.01), indicating greater attachment insecurity compared to controls. Additionally, relationship quality, as assessed by the Dyadic Adjustment Scale (DAS), was significantly lower in the vulvodynia group (t(202) = 9.60, *p* < 0.001, Cohen’s d = 1.34).

Effect sizes were particularly large for sexual dysfunction (Cohen’s d = 0.98), depressive symptoms (Cohen’s d = 2.22), and attachment avoidance (Cohen’s d = 2.01), highlighting the profound impact of vulvodynia on these domains. These findings underscore the complex interplay between physical symptoms, psychological distress, and relational challenges in women with vulvodynia.

## 4. Discussion

The results of this study highlight significant differences between the vulvodynia group and the control group across various psychological, relational, and fertility-related domains. This section discusses the main findings, focusing on the observed differences and their clinical implications.

### 4.1. Sociodemographic Characteristics

This study provides important insights into the psychological aspects of women with vulvodynia, suggesting significant differences compared to the control group in terms of marital status, living situation, and relationship type for the sociodemographic characteristics.

The observed higher prevalence of separation and divorce among women with vulvodynia compared to the control group aligns with the previous literature on chronic pain conditions and their impact on intimate relationships. Chronic pain, particularly when associated with sexual dysfunction, has been linked to relational distress and dissatisfaction, often contributing to increased rates of separation and divorce. In the context of vulvodynia, dyspareunia and the resulting avoidance of sexual intimacy can create significant strain within romantic partnerships, leading to reduced emotional closeness and increased relational conflict [29,30]. These findings are consistent with research demonstrating that sexual dysfunction and chronic pain conditions often disrupt relational dynamics, contributing to heightened emotional distress and dissatisfaction [31,32]. Furthermore, the significantly lower rates of engagement and cohabitation among women with vulvodynia suggest that the condition may also hinder the formation of stable romantic relationships, potentially due to fears of intimacy, pain-related anxiety, and perceived inadequacy in sexual relationships [33].

The differences in living situations observed between the two groups may also reflect the broader social impact of vulvodynia. Women with vulvodynia were more likely to live with family members or alone and less likely to cohabit with a partner compared to the controls. This pattern indicates the increased reliance on familial support often observed in individuals with chronic health conditions [34]. The physical and emotional burden of vulvodynia may necessitate greater practical assistance and emotional care, leading women to maintain close ties with their families of origin. Alternatively, the higher rates of living alone in the vulvodynia group could reflect social withdrawal and isolation, phenomena frequently documented in individuals experiencing chronic pain [35]. Social withdrawal can result from both physical limitations and the psychological distress associated with persistent pain, including anxiety, depression, and reduced self-efficacy in social interactions. These findings suggest the need for targeted interventions aimed at enhancing social support networks and mitigating the risk of isolation among women with vulvodynia.

### 4.2. Clinical Aspects and Implications

The clinical findings of this study shed light on the profound psychological and relational challenges faced by women with vulvodynia, highlighting significant differences across all psychometric measures compared to the control group. Women with vulvodynia reported elevated levels of attachment anxiety and avoidance, heightened depressive symptoms, lower resilience, and greater fertility-related distress. The strong association between vulvodynia and psychological distress aligns with previous research indicating that chronic pain conditions often exacerbate symptoms of anxiety, depression, and emotional dysregulation [36,37]. This interplay suggests that the persistence and severity of vulvodynia symptoms may be influenced by underlying psychological vulnerabilities, such as attachment insecurities and maladaptive coping strategies [38]. The observed lower scores on dyadic adjustment and higher couple relationship concerns reflect the strain that chronic pain places on intimate partnerships. This study demonstrates that the emotional withdrawal associated with avoidant attachment and the heightened dependency driven by attachment anxiety can both contribute to relational dissatisfaction and increased conflict.

#### 4.2.1. Bidirectional Relationship Between Vulvodynia and Psychological Factors

The relationship between vulvodynia and psychological and behavioral patterns is bidirectional. On one hand, the chronic pain associated with vulvodynia can exacerbate attachment insecurity, anxiety, and avoidance behaviors. On the other hand, pre-existing psychological vulnerabilities, such as attachment anxiety or high levels of stress, may predispose women to develop vulvodynia or worsen its symptoms. Recent studies suggest that chronic stress, fear, and anxiety can activate the pelvic stress reflex, leading to increased pelvic floor muscle tension and potentially contributing to the onset or persistence of vulvodynia [39]. This understanding reinforces the importance of integrated therapeutic approaches that address both psychological and physiological dimensions in women with vulvodynia.

#### 4.2.2. Attachment Styles

In this study, the significantly higher levels of attachment anxiety and avoidance observed in the vulvodynia group, as measured by the Experience in Close Relationships Scale (ECR-12), indicate greater attachment insecurity compared to the controls. These findings align with prior research suggesting that chronic pain conditions can intensify attachment-related distress, leading to heightened fears of abandonment and difficulties with emotional intimacy [40]. Studies have shown that individuals with anxious or avoidant attachment patterns often experience heightened pain perception and increased psychological comorbidities, such as depression and anxiety. Additionally, attachment insecurity has been linked to increased psychological distress in individuals with chronic pain, suggesting that these relational patterns may contribute to the persistence and severity of vulvodynia symptoms [41]. These psychological factors are prevalent among women with vulvodynia and may exacerbate physical symptoms, creating a feedback loop of distress. Furthermore, attachment insecurity has been associated with greater difficulties in emotional regulation and coping strategies, which can intensify the perception of pain and related distress [42]. Avoidant attachment often leads to emotional withdrawal and reduced communication, while attachment anxiety may result in heightened dependency and fear of rejection. Both patterns can contribute to relational dissatisfaction and increased conflict, thereby compounding the psychological burden of the condition [43].

#### 4.2.3. Temperament

The results of the Approach–Avoidance Temperament Questionnaire (ATQ) revealed significant differences in both the approach and avoidance dimensions between the vulvodynia group and the control group. Women with vulvodynia demonstrated lower approach temperament and higher avoidance temperament, indicating a reduced tendency to engage in rewarding experiences and an increased sensitivity to potential harm or threat. Recent studies have highlighted the critical role of temperament in shaping psychological and relational outcomes in individuals with chronic pain conditions. Specifically, a high avoidance temperament has been associated with heightened pain-related fear and catastrophizing, which are strongly linked to increased pain perception and disability [44].

Conversely, a lower approach temperament reflects a reduced inclination toward seeking positive experiences and engaging in goal-directed behaviors. This lack of approach motivation has been associated with diminished psychological well-being and lower resilience, further complicating the management of chronic pain [45,46,47,48,49]. Within the context of vulvodynia, low approach temperaments may hinder women’s ability to pursue fulfilling personal and relational goals, thereby perpetuating a cycle of avoidance and dissatisfaction.

#### 4.2.4. Resilience

Resilience is a multidimensional construct determined by the interplay among biological, emotional, and external factors.

Resilience appears to play a critical role in coping with chronic pain and its psychological consequences [50]. Our findings revealed significantly lower resilience scores on the Resilience Scale (RS-14) in the vulvodynia group compared to the controls, indicating a reduced capacity for coping with stress and adversity. This diminished resilience may exacerbate the psychological burden associated with vulvodynia, including depressive symptoms, anxiety, and relational distress [51,52].

Resilience also involves specific neurobiological mechanisms, which include modifications in neurotransmitter levels, increased and prolonged blood levels of glucocorticoids and cortisol, and elevated inflammatory cytokine production [53].

Vulvodynia, as other chronic pain conditions, is linked to changes in neural plasticity and connectivity across various brain regions, such as the hippocampus, the locus coeruleus/norepinephrine system, the anterior cingulate cortex, the prefrontal cortex, the thalamus, the cerebellum, the periaqueductal gray matter, the mesolimbic reward network, and the fear circuitry. The shared involvement of these areas in both pain perception and mood regulation provides insight into why individuals with chronic pain and depression often experience vulvar pain and other somatic symptoms.

In our study, all participants with vulvodynia demonstrated poor resilience: the activation of emotional and limbic brain regions in women with persistent painful conditions may underlie the process of pain chronicization [54].

#### 4.2.5. Fertility-Related Stress

The results using the Fertility Problem Inventory—Short Form (FPI-SF) revealed significant differences between groups in social concern and couple relationship concern, highlighting the fertility-related distress experienced by women with vulvodynia. The impact of chronic pain on sexual function and intimacy can exacerbate concerns about fertility and reproductive planning, as previously documented in the literature on dyspareunia and chronic pelvic pain [55,56]. Despite the prevalence of vulvodynia, its impact on fertility-related stress remains underexplored. Women may experience fear and avoidance of sexual activities, resulting in decreased libido and intimacy issues within relationships. The psychological burden of anticipating pain during intercourse can further diminish sexual desire and satisfaction, creating a cycle of pain and sexual avoidance. This cycle not only affects the individual’s sexual health but also places strain on partner relationships, potentially leading to increased fertility-related stress.

The distress associated with potential infertility can lead to heightened anxiety, depression, and social isolation. Moreover, societal and cultural pressures to conceive may exacerbate these psychosocial stressors, further impacting the mental health and well-being of affected women. Despite the challenges posed by vulvodynia, studies suggest that the condition has minimal effect on the decision to conceive. Research indicates that women with more intense or unpleasant pain do not avoid or fear pregnancy more than those with less pain [57].

However, the journey to conception may be fraught with difficulties due to pain-related sexual dysfunction, necessitating the use of assisted reproductive technologies or alternative conception methods. Healthcare providers should be cognizant of these challenges and offer appropriate guidance and support to women navigating reproductive decisions while managing vulvodynia [58].

#### 4.2.6. Sexual Functioning

In this study, the Female Sexual Function Index (FSFI) demonstrated significant impairments across all domains of sexual function, including desire, arousal, lubrication, orgasm, and satisfaction, in the vulvodynia group. These deficits are not only statistically significant but also clinically meaningful, reflecting substantial challenges in sexual health and intimate relationships [59,60].

These results are consistent with research documenting the severe sexual dysfunction associated with vulvodynia, which often contributes to relational distress and reduced quality of life. Several studies consistently highlight the high prevalence of sexual dysfunction among women with vulvodynia, demonstrating a clear association between pain severity, psychological distress, and sexual difficulties [61,62]. Women with vulvodynia often report markedly reduced sexual satisfaction, which is closely linked to heightened pain levels, disruptions in daily activities due to pain, and comorbid symptoms of depression and anxiety [63].

Sexual dysfunction in women suffering from vulvodynia appears to affect all domains of sexual functioning. These include diminished sexual desire, pain during intercourse (dyspareunia), and difficulties with arousal and orgasm [64,65].

#### 4.2.7. Relationship Quality

The study findings, indicating lower scores on the Dyadic Adjustment Scale (DAS-7) among women with vulvodynia, underscore the significant relational distress and diminished relationship satisfaction prevalent in this population. This observation is consistent with the existing literature, which highlights the profound impact of chronic pain conditions on romantic relationships. These findings align with previous studies showing that chronic pain conditions often disrupt romantic relationships, leading to increased conflict, reduced emotional closeness, and higher rates of separation and divorce [66].

Chronic pain, particularly vulvodynia, has been shown to disrupt intimate partnerships by increasing conflict, reducing emotional closeness, and elevating the risk of separation and divorce. The persistent nature of vulvar pain can lead to avoidance of sexual activity, fostering feelings of frustration and misunderstanding between partners. This dynamic often results in a negative feedback loop, where pain exacerbates relational strain, which in turn may intensify the perception of pain. Empirical studies have demonstrated that male partners’ responses to painful intercourse significantly influence women’s pain experiences and sexual satisfaction. Solicitous or overly protective behaviors, while well-intentioned, can inadvertently reinforce pain behaviors and contribute to increased pain perception. Conversely, facilitative responses that encourage adaptive coping are associated with improved sexual function and relationship satisfaction [67].

Furthermore, research indicates that couples’ communication patterns play a crucial role in moderating the impact of vulvodynia on relationship quality. Effective communication, characterized by validation and empathy, is associated with lower pain intensity and better dyadic adjustment. In contrast, invalidating communication patterns correlates with heightened pain and relational dissatisfaction [68].

The interplay between vulvodynia and relationship quality is complex and bidirectional. Given these insights, addressing relationship quality through targeted interventions emerges as a critical component in the management of vulvodynia. Couples counseling that focuses on enhancing communication skills, fostering emotional intimacy, and developing effective pain-coping strategies can mitigate relational distress. Additionally, incorporating sexual therapy to address specific sexual dysfunctions associated with vulvodynia may further improve relational and individual well-being [69].

#### 4.2.8. Depressive Symptoms

The markedly elevated depressive symptoms observed in this study in women with vulvodynia, as measured by the Beck Depression Inventory (BDI), highlight the considerable psychological burden associated with this chronic pain condition. This finding is consistent with extensive research indicating a strong correlation between chronic pain and depression, wherein persistent physical discomfort, social isolation, and relational distress contribute to the development and exacerbation of depressive symptoms [70,71].

The bidirectional relationship between chronic pain and depression suggests that each condition can potentiate the other, creating a cycle that complicates treatment and diminishes quality of life. In the context of vulvodynia, the chronic nature of vulvar pain can lead to significant emotional distress, impacting daily functioning and interpersonal relationships. This underscores the necessity for comprehensive treatment plans that address both the physical and psychological aspects of the condition [72].

### 4.3. Clinical Implications

The clinical findings of this study highlight the complex interplay between psychological, emotional, relational, and sexual factors in women with vulvodynia. The significant impairments observed across all psychometric measures emphasize the need for a comprehensive, multidisciplinary treatment approach that addresses the condition’s medical, psychological, and social dimensions. Future research should explore the long-term impact of these factors on treatment outcomes and quality of life, as well as the effectiveness of integrated intervention models in improving the well-being of women with vulvodynia [73,74].

## 5. Conclusions

### 5.1. Conclusions and Policy Implications

These findings highlight the potential psychosocial burden of vulvodynia, emphasizing the need for healthcare providers to address not only the physical symptoms but also the relational and emotional consequences of the condition. Given that relationship dissatisfaction and lack of social support are associated with poorer pain outcomes and higher psychological distress [75,76], interventions aimed at improving partner communication, providing psychological counseling, and fostering social support networks could be beneficial for women with vulvodynia.

Overall, the significant differences observed in marital status, relationship type, and living situation between the vulvodynia and control groups suggest that vulvodynia may have profound consequences for relational and social well-being. These results, together with the substantial psychological distress, impaired sexual functioning, and lower resilience documented in the clinical findings, underscore the importance of a multidisciplinary approach to vulvodynia treatment.

The sociodemographic disparities identified in this study emphasize the necessity of going beyond the management of physical symptoms. Healthcare providers should prioritize the social and relational dimensions of vulvodynia, offering couples counseling, psychoeducation, and tailored social support interventions. The high levels of attachment anxiety and avoidance, reduced resilience, and severe sexual dysfunction observed in women with vulvodynia highlight the need for psychological therapies focused on enhancing emotional regulation, fostering secure attachment patterns, and improving intimacy within romantic relationships.

In addition, addressing fertility-related concerns is vital, given the elevated distress reported in this domain. The combination of chronic pain and anxiety related to reproductive health calls for closer collaboration between gynecologists, psychologists, and fertility specialists. Future policy efforts should focus on increasing access to specialized care for vulvodynia, reducing diagnostic delays, and combating the stigma often associated with this condition. Developing standardized Diagnostic Therapeutic Care Pathways (PDTAs) and including vulvodynia in Essential Levels of Care (LEA) across more regions in Italy can facilitate timely and effective interventions, ultimately improving patient outcomes [7,77,78,79].

### 5.2. Limitations and Recommendations for Future Research

This study acknowledges several limitations that could affect the generalizability and depth of its findings. First, the sample is limited to Italian-speaking women, which restricts cultural applicability and makes it difficult to generalize the results across diverse populations. Second, the cross-sectional design prevents exploring causal relationships between psychological, relational, and clinical variables. Third, while this study includes validated psychometric measures, the self-report nature of the data could introduce response biases, such as social desirability or subjective misinterpretation of items.

Additionally, this study does not account for potential confounding variables like vulvodynia’s duration and treatment history, which could influence psychological and relational outcomes. It is important to acknowledge that, despite efforts to standardize the diagnostic process, the subjective nature of vulvodynia diagnosis—often based on the exclusion of other conditions—may have introduced some heterogeneity in the criteria applied by different gynecologists. This limitation should be further explored in future research. Additionally, the use of convenience sampling for the control group may introduce selection bias, even though participants were carefully selected based on clear inclusion and exclusion criteria. To address this concern, we compared key covariates (e.g., age, education level, and marital status) between the two groups and found minimal differences, indicating a good balance.While we recognize that advanced techniques such as propensity score matching (PSM) could further enhance comparability, PSM was not implemented in this study due to the structured recruitment process for the control group and limitations in sample size and available covariates, which may have hindered the feasibility of reliable matching. Future studies with larger datasets are encouraged to explore PSM or other advanced matching techniques to further reduce selection bias and improve the robustness of the findings.

To further enhance the generalizability and applicability of these findings, future studies should prioritize the following issues:

1. Diverse Populations: Expand research to include clinical samples, such as individuals undergoing fertility treatments or coping with infertility, and cross-cultural comparisons to evaluate the adaptability of this study across different societal contexts. Exploring the intersection of vulvodynia with other chronic conditions and cultural attitudes toward sexual health would provide a more nuanced understanding of the condition.

2. Longitudinal Approaches: Conduct longitudinal studies to explore how psychological and relational aspects related to vulvodynia evolve over time. Investigating the long-term impact of attachment insecurity, resilience, and sexual dysfunction on relationship stability and quality of life could reveal important trends and inform more effective intervention strategies.

3. Intervention Effectiveness: Evaluate the efficacy of multidisciplinary treatment models combining medical, psychological, and relational therapies. Comparative studies on different therapeutic approaches, including cognitive-behavioral therapy, mindfulness-based interventions, and couples therapy, would help identify best practices for addressing the complex needs of women with vulvodynia.

4. Sociocultural Factors: Investigate the role of sociocultural influences in shaping the experiences of women with vulvodynia, particularly in relation to stigma, medical misdiagnosis, and access to specialized care. Understanding these factors can guide the development of culturally sensitive treatment programs and public health initiatives to raise awareness and promote early diagnosis.

In conclusion, this study highlights the profound impact of vulvodynia on the psychological, relational, and social well-being of affected women. By adopting a holistic, multidisciplinary approach to treatment and expanding research efforts to address existing knowledge gaps, healthcare providers and policymakers can work toward improving care pathways and enhancing the quality of life for women living with this debilitating condition.

## Figures and Tables

**Table 1 ijerph-22-00527-t001:** Sociodemographic characteristics of the vulvodynia and control groups (N = 203).

Variable	Category	Vulvodynia (*n* = 96)	Control (*n* = 107)	χ^2^ (df)	*p*
**Education**	High school diploma or vocational degree	44 (45.8%)	37 (34.6%)	2.687 (2)	0.261
	University degree or higher	50 (52.1%)	67 (62.6%)		
	Middle school diploma	2 (2.1%)	3 (2.8%)		
**Marital status**	Married	21 (21.9%)	27 (25.2%)	15.892 (5)	0.007 *
	Cohabiting	3 (3.1%)	17 (15.9%)		
	Single	57 (59.4%)	58 (54.2%)		
	Separated or divorced	12 (12.5%)	5 (4.7%)		
	Widowed	1 (1.0%)	0 (0.0%)		
**Living situation**	Other	0 (0.0%)	1 (0.9%)	9.920 (4)	0.042 *
	With other family members or friends	45 (46.9%)	38 (35.5%)		
	With partner	39 (40.6%)	63 (58.9%)		
	Alone	11 (11.5%)	5 (4.7%)		
	With children	1 (1.0%)	0 (0.0%)		
**Sexual orientation**	Bisexual	2 (2.1%)	5 (4.7%)	1.281 (2)	0.527
	Heterosexual	93 (96.9%)	100 (93.5%)		
	Homosexual	1 (1.0%)	2 (1.9%)		
**Relationship type**	Cohabiting	5 (5.2%)	6 (5.6%)	15.467 (6)	0.017 *
	Common law partnership	0 (0.0%)	1 (0.9%)		
	Engaged	57 (59.4%)	72 (67.3%)		
	In a relationship	5 (5.2%)	0 (0.0%)		
	Single/free	0 (0.0%)	1 (0.9%)		
	Separated or divorced	9 (9.4%)	1 (0.9%)		
	Married	20 (20.8%)	26 (24.3%)		
**Employment status**	Housewife	3 (3.1%)	1 (0.9%)	8.338 (6)	0.214
	Unemployed/seeking first job	9 (9.4%)	2 (1.9%)		
	Self-employed	22 (22.9%)	31 (29.0%)		
	Temporary contract worker	13 (13.5%)	13 (12.1%)		
	Permanent contract worker	30 (31.3%)	36 (33.6%)		
	Retired	0 (0.0%)	1 (0.9%)		
	Student	19 (19.8%)	23 (21.5%)		
**Income level**	Low	64 (66.7%)	74 (69.2%)	2.752 (2)	0.253
	Medium	23 (24.0%)	29 (27.1%)		
	High	9 (9.4%)	4 (3.7%)		
**Children**	No	65 (67.7%)	71 (66.4%)	3.415 (4)	0.491
	Yes, 1	19 (19.8%)	21 (19.6%)		
	Yes, 2	8 (8.3%)	14 (13.1%)		
	Yes, more than 2	3 (3.1%)	1 (0.9%)		

**Note**: * *p* < 0.05 indicates significant differences between groups.

**Table 2 ijerph-22-00527-t002:** Comparison of psychological and clinical variables between groups (N = 203).

Measure	Vulvodynia (*n* = 96)	Control (*n* = 107)	t (df)	*p*	Cohen’s D
ECR-12					
Attachment Anxiety	29.20 (5.57)	23.61 (5.77)	6.96 (198)	<0.001	0.98
Attachment Avoidance	26.50 (5.03)	16.90 (4.54)	14.27 (200)	<0.001	2.01
ATQ					
Approach	28.43 (3.72)	30.65 (3.86)	−4.16 (200)	<0.001	−0.58
Avoidance	31.11 (3.7)	23.21 (4.32)	13.83 (199)	<0.001	1.95
RS-14					
Total Score	69.97 (8.67)	78.85 (9.85)	−6.74 (199)	<0.001	0.95
FPI-SF					
Social Concern	36.39 (6.09)	27.75 (6.57)	9.60 (197)	<0.001	0.319
Need for Parenthood	19.07 (7.65)	20.36 (7.96)	−1.17 (201)	0.12	
Rejection of a Child-Free Lifestyle	22.69 (5.64)	21.44 (5.86)	1.53 (200)	0.06	
Couple Relationship Concern	15.08 (4.89)	9.58 (4.42)	8.39 (199)	<0.001	1.18
Total Score	93.17 (13.18)	79.22 (14.05)	7.18 (196)	<0.001	1.02
FSFI					
Desire	3.69 (1.35)	4.97 (0.79)	−8.11 (201)	<0.001	1.16
Arousal	3.38 (1.89)	5.21 (0.80)	−8.79 (201)	<0.001	1.28
Lubrification	3.16 (1.87)	5.41 (0.84)	−10.83 (200)	<0.001	−1.57
Orgasm	3.12 (2.05)	5.32 (0.87)	−9.74 (200)	<0.001	1.43
Satisfaction	2.98 (1.94)	5.30 (0.88)	−10.74 (201)	<0.001	−1.56
Pain	2.52 (1.63)	5.18 (1.06)	−13.57 (201)	<0.001	1.94
Total Score	18.77 (9.19)	31.42 (4.04)	−12.38 (199)	<0.001	1.81
DAS-7					
Total Score	24.48 (4.78)	31.71 (4.73)	−10.81 (201)	<0.001	1.51
BDI					
Total Score	16.69 (7.39)	4.34 (3.14)	15.04 (199)	<0.001	2.22

## Data Availability

The data used in this study are from the Dipartimento di Scienze della Salute (DSS), Università degli Studi di Firenze, Via San Salvi 12, 50135 Firenze. The data generated and/or analyzed as part of this study are available from the corresponding author upon request.

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
